# The Value of ‘Cow Signs’ in the Assessment of the Quality of Nutrition on Dairy Farms

**DOI:** 10.3390/ani12111352

**Published:** 2022-05-25

**Authors:** Kiro Risto Petrovski, Paul Cusack, Jakob Malmo, Peter Cockcroft

**Affiliations:** 1Davies Livestock Research Centre, School of Animal and Veterinary Sciences, The University of Adelaide, Roseworthy Campus, Roseworthy 5371, Australia; 2Australian Livestock Production Services, Cowra 2794, Australia; alpscowdr@bigpond.com; 3MAFFRA Veterinary Centre, Maffra 3860, Australia; jmalmo@maffravet.com.au; 4School of Veterinary Medicine, University of Surrey, Guilford, Surrey GU2 7AL, UK; p.cockcroft@surrey.ac.uk

**Keywords:** appetite, demeanor, fecal digestibility scoring, fecal scoring, fecal perineal staining, obtundancy, prehension, rumen fill scoring, rumination, thirst

## Abstract

**Simple Summary:**

Cow signs are behavioral, physiological, and management parameters that can be observed and measured. Cow signs can be used as a field approach to evaluate the composition of the ration, the quality of rumen fermentation, the quality of digestion, and the general herd health of cattle of interest. This review of cow signs associated with nutrition provides farm advisors, consultants, nutritionists, practitioners, and dairy farmers with an additional toolkit that can be used to improve the assessment of the quality of dairy cattle nutrition. ‘Cow signs’ are not to be used alone as a sole tool for assessment of the quality or nutrition of dairy cows. Some of the ‘cow signs’ are incorporated in precision technologies on many dairy farms and are extensively used in the assessment of dairy cow welfare, health, and nutrition.

**Abstract:**

The aim of this review is to provide dairy farm advisors, consultants, nutritionists, practitioners, and their dairy farmer clients with an additional toolkit that can be used in the assessment of the quality of their dairy cattle nutrition. Cow signs are behavioral, physiological, and management parameters that can be observed and measured. They are detected by examining and observing the cattle. Other physiological parameters such as fecal scoring, rumen fill, and body condition scoring are also included in ‘cow signs’. The assessment should be both qualitative and quantitative; for example, is the cattle individual lame and what is the severity of lameness. The ‘diagnosis’ of a problem should be based on establishing a farm profile of ‘cow signs’ and other relevant information. Information gathered through assessment of cow signs should be used as an advisory tool to assist and improve decision making. Cow signs can be used as part of an investigation and or farm audit.

## 1. Introduction

Dairy cattle productivity, health, and fertility are significantly affected by the quality and quantity of the nutrition, making dairy cattle nutrition an important profit driver [1]. The biggest issue when feeding dairy cattle is addressing both metabolic compartments, namely, the rumen microbes and the bovine tissues [2,3]. Therefore, the nutrition should aim to keep the rumen functioning optimally whilst providing the appropriate levels of nutrients for health, maintenance, and production [4]. The assessment of the quality and quantity of dairy cattle nutrition is a critical task for dairy herd practitioners, consultants, and dairy farmers/managers. The term practitioners in this article means a herd-level advisor or consultant, nutritionist, or veterinary practitioner.

A variety of approaches to assess the adequacy of nutrition in dairy cattle have been adopted by veterinarians in practice or consultancy services. The majority of practitioners use a combination of methods, including assessment of the value of production indices and cow sign profiles, utilization of nutritional analysis of the diet, and/or metabolic profiling [5]. The most commonly used method of assessment of the adequacy of nutrition is the feed (nutrition) analysis of a representative sample of the diet/feedstuffs. Significant limitations to this approach are the uncertainty regarding how representative the sample is and the cost.

Additionally, nutritional analysis does not consider the differences between the calculated diet, the diet that is actually prepared and offered, and the diet eaten due to sorting and quantity supplied [5,6,7,8]. The quantity and quality of nutrition provided by the diet to an individual dairy cow can only be estimated. It is affected by a variety of factors, including feeding space availability, social hierarchy, access to high-quality drinking water, and the health of a particular individual dairy cow [9,10]. An additional approach is to use parameters that measure the response to the diet offered, considering that the ultimate judge of the quality and quantity of nutrition are the end users—dairy cattle [5,6]. These parameters, which are more commonly known as cow signs or cow signals, have the potential to be of value irrespective of environmental conditions [11], genetic background [11], production status, age, or size of the dairy cattle of interest.

Cow signs [1,12] are cattle behavioral, physiological, and management parameters that can be observed and measured [11,13,14,15,16,17]. Observation of cow signs does not require specialized equipment and laboratory analyses [1,11]. Additionally, cattle-based parameters included in the profiling of cow signs are independent of (but correlate with) the production parameters, and do not require sophisticated software to be analyzed and interpreted. The important cow signs associated with nutrition include those related to the general condition of cattle (e.g., body condition score, demeanor, and hair coat characteristics), behavioral signs associated with feeding and drinking (e.g., appetite, prehension, rumination, and thirst), and physiological parameters related to nutrition (e.g., fecal digestibility score, fecal score, and rumen fill) [1,5]. Some authors add the assessment of the ‘stretch and scratch’ factor of the diet as part of cow signs [5] while others do not. Cow signs can be used as a field approach to evaluate the composition of the ration, the quality of rumen fermentation, the quality of digestion, and the general herd health of cattle of interest [4,18].

During the assessment of cow signs, the practitioner should be as least disruptive as possible on the comfort of the group of cattle being assessed. Animal-based indicators are also becoming increasingly important as an assessment tool for animal welfare, in addition to resource-based indicators (e.g., bedding, hygiene, management, and quality of buildings) [11,15,16,17,19]. The assessment should be both quantitative, with scoring of the deviations of signs from a predetermined baseline (e.g., the rumen fill 2 h after access to food should be above 2.5), and qualitative, involving assessment of the severity of the deviation from ‘normal’ (e.g., the rumen fill is acceptable or non-acceptable). An evidence-based understanding of the conditions and disorders that can cause deviations from normal of a particular sign is crucial for the correct interpretation. The prevalence and ‘severity’ or the deviation from the normal of the cow signs at the group/herd level provides a profile that can assist in the diagnosis of the problem [1,5,12,20].

## 2. Limitations of ‘Cow Signs’

The use of cow signs on a dairy farm has indeed some limitations. Firstly, recognition of many of the ‘cow signs’ discussed in this review requires experience and attention to detail. Secondly, on many dairy enterprises, a significant impediment to implementation may be the time demands to correctly identify and utilize the ‘cow signs’. Thirdly, many dairy enterprises will lack the essential information for implementing ‘cattle signs’ such as daily individual cow milk fat percentage. Furthermore, some ‘cow signs’ are affected by the stage of lactation and physiological changes in nutrition over lactation, such as appetite and rumen fill. Finally, some ‘cow signs’ are affected by the reproductive stage of the cow, such as rumen fill in heavily pregnant cows.

Therefore, ‘cow signs’ should never be used as a sole tool for assessment of the quality of nutrition. In this review, we did not discuss routinely collected herd data, such as body condition scoring, fertility indexes, milk composition and quantity, and nutritional analysis records. Their potential value in estimating nutrition is recognized, but we felt they are already discussed elsewhere. The discussion in this review emphasizes individual cattle observations but ‘acceptable norms’ at the population level are provided where applicable. We hope that this material also can be used for training of future practitioners in examination of the alimentary system.

## 3. Behavioral Parameters

### Assessment of Demeanor

As a prey species, the olfactory and auditory functions of cattle are well developed. Conversely, their vision is less well developed. Appraisal of cattle usually begins with the assessment of demeanor. Altered demeanor can present as obtundancy (depression) or excitement. The assessment should be made with awareness that the general behavioral response of cattle towards humans depends upon their previous exposure and experiences (e.g., gently handled dairy cattle are calmer around humans).

Obtundancy is characterized by a decreased interest in their surroundings and diminished responsiveness to external stimuli. Obtundancy can be a manifestation of a wide range of conditions involving a variety of body systems. Obtundancy of dairy cattle at the population or group level may relate to feeding (e.g., a sign of lactic acid accumulation in the rumen) [6,7,21]. The depth of depression is usually correlated with the severity of the underlying disorder.

Hyperreactivity (excitement) in individual dairy cattle may be easily overlooked, particularly in its milder forms. Hyperreactivity is the abnormally heightened reaction to a stimulus. Responses vary widely from mildly overreacting (excitement) to bellowing and fractious behavior towards herd mates and people (frenzy). Hyperreactive behaviors can include hyperesthesia, excitement or apprehension, restlessness, mania, and frenzy. Hyperexcitability of dairy cattle at the population and group level may indicate mishandling due to being startled or may be due to nutritional factors (e.g., subclinical to mild clinical hypomagnesemia). The severity of obtundancy/excitement can be scored (see Table 1 and Table 2 [22,23,24,25,26,27,28]) using an attitude score on a scale of 1 to 5. An attitude score of one relates to a bright and alert dairy cattle individual with no signs of depression or excitement. The scores 2 through 5 represent the progression of depression or excitement from mild, moderate, to severe.

Dairy cattle may undergo progressive changes in demeanor from Score 1 to Score 5 (for either depression or excitement), usually occurring gradually over a few days, indicating an increase in the severity of the condition. Hence, an observant client may have time to take action (e.g., correct acidosis or hypomagnesemia). The decision to take action is best guided by the progressive increase in the score. Regular recording and data assessment allow for timely detection of the progression.

## 4. Assessment of the Skin and Hair Coat

The condition of the skin and coat can give an indication of the nutritional status; however, practitioners should be aware that other conditions can also affect this system. The general appearance of the skin and hair coat, including the absence of lick marks (e.g., acidosis), dullness of the hair coat (e.g., undernutrition, several deficiencies in macro- and micronutrients), discoloration (e.g., copper deficiency), and overall clinical impression (e.g., undernutrition, imbalanced nutrition, zinc deficiency, ingestion of photosensitization agents), may be indicative of nutritional mismanagement but are difficult to interpret [16,29].

### Fecal Perineal Staining

Many digestive problems of dairy cattle are characterized by lower fecal scores with increased water content. Feces with increased water content are typically visible by fecal staining of the perineal area. A scoring system devised by the first author is presented in Table 3. Fecal perineal staining can also be caused by infectious diarrhea (e.g., gastro-intestinal parasitism, Johne’s disease), disorders of other body systems (e.g., congestive heart failure, chronic amyloidosis), or environmental conditions (e.g., sudden weather change) [30]. Fecal perineal staining can be used as an indirect indicator of nutritional problems (e.g., subacute ruminal acidosis) but other etiologies must be considered [6,7,18,31,32,33,34,35,36,37]. It should be noted that dairy cattle grazing lush, high-quality pasture will commonly have lower fecal scores and increased fecal staining of the perineal area (fecal staining score of 3–4 may be ‘normal’).

## 5. Assessment of Feeding

The feeding assessment is very important and may provide useful insights into the feed quality, quantity, and delivery method. Important cow signs related to feeding include appetite, thirst, prehension, and rumination. Aberrant feeding behavior may be present in the whole group of dairy cattle or only in an individual. Nutrition-related aberrant feeding behavior is usually present in the majority or the whole group. Conversely, aberrant feeding behavior in only one or few dairy cattle is more likely to be related to clinical conditions affecting the individual.

### 5.1. Appetite

Appetite is the desire to eat the offered feedstuffs and is mainly assessed through the feed intake. Feed intake can be affected by a number of factors, including climatic conditions, diet composition, social dynamics, feed availability, delivery systems used, body size, stage and level of production, pregnancy, age of cattle, stress, and exercise [2,7,9,38,39,40,41,42,43,44,45]. The accepted knowledge of the effect of breed and body size on the utilization of feed by dairy cattle has been questioned by newer publications [46,47,48,49]. Previously, it was accepted that dairy cattle of the Jersey breed and individual cattle of larger frame and weight eat more per unit of body weight [49].

The appetite can be normal, decreased (inappetence, hypophagia), increased (polyphagia), abnormal (alotriophagia), or completely absent (anorexia). Aberrant appetite can be temporary or permanent. Medical causes of depressed appetite in dairy cattle include a lack of desire for food or an inability to prehend, masticate, and/or swallow [50]. Environmental causes of depressed appetite include temperature extremes and heat stress [51]. The lack of desire for food from a nutritional aspect can be caused by acidosis [6,14,18,21,32,33,34,35,36,37,49,52,53]. Causes of depressed appetite due to factors directly related to the feed provided include low palatability (e.g., offensive smell and taste, rough appearance, inappropriate texture and feel), the presence of decomposition, endophytes, mold or mycotoxins [2,5,7,39,40,41,51,54,55], or unfamiliarity with the offered diet [39,56]. Additionally, total dry matter intake per day (i.e., the appetite) can be affected by some physico-chemical properties of the diet, such as the fiber type and length, and content (e.g., neutral detergent fiber, NDF), digestibility of other carbohydrates, fat content of the diet, particle size, particle fragility, diet weight, rumen degradation and fermentation, passage rate through the digestive system, osmolarity of the rumeno-reticulum, and production of ruminal degradation products (e.g., concentration of various volatile fatty and other simple acids) [6,9,35,38,39,40,41,42,49,51,57,58,59,60,61,62,63,64,65,66,67,68,69,70,71,72,73,74]. For example, excess physical neutral detergent fiber in the diet can result in physical limitations caused by distention of various portions of the digestive tract (predominantly rumen and reticulum) [38,39,40,64,71,75,76,77,78,79]. Distension of the digestive system results in stimulation of satiety receptors [38,40,78,80]. In contrast, decreased appetite without distention of various portions of the digestive system may result from excessive availability of easy-digestible carbohydrates associated with low ruminal pH [35,38,40,53]. Finally, insufficient fiber in the diet also increases the risk of stereotypic behavior in housed cattle [81].

The appetite of cattle can be affected by delivery methods and access factors [5,7,9,39,50]. Generally, dairy cattle that have no access to food for some time, including pasture, have a good appetite when offered the usual diet [82,83]. However, the provision of food alone does not ensure good intake because appetite may be significantly influenced by the delivery method. Thus, the design and maintenance of the feeding facilities are also important. Access to feeding areas may be affected by available feeding space, social interactions/hierarchies, quality of the bunker/feedpad/ pasture surfaces, and hygiene [5,6,9,10,50,84,85]. A good ‘rule of thumb’ is to provide 120% feeding spaces for the number of cattle with 60–85 cm per feeding space. The practitioner should be aware that with greater body size the feeding space requirements increase. Availability and sorting of food is assessed by observing the residual food in the bunker/feedpad/paddock after each feeding period. The proportion of residual food in the bunker/feedpad should be no more than 3% to 4% at the end of the prescribed feeding period, just before a new batch of feed is deposited [58]. For female cattle in the transition period, the proportion of residual food may be as high as 15%. In pasture-based systems, the residual food is usually estimated as ‘residual herbage mass’; i.e., the total weight of herbage per unit area, measured after grazing to a ground level [86]. Grazing usually satisfies dairy cattle requirements provided the sward height does not drop below 8–10 cm. The feeding efficiency in pasture-based cattle decreases when the herbage mass falls below 2000 kg/ha as the bite size decreases and is offset by an increase in the required grazing times [87,88,89]. The residual herbal mass is usually assessed at the end of the grazing period just before the dairy cattle are moved to a new break/paddock.

The forage material that is less palatable, spoiled, or of a poorer quality than the rest can be sorted out by cattle during feeding and rejected [58,63,90], depending on mixing and feed allocation. This feed is of a lower digestibility or reduced palatability. If consumed, it will likely reduce the feed intake and ultimately lead to lowered productivity. Therefore, dairy cattle that are being forced to eat the residual feed left in the bunker/feedpad/pasture may be underfed as they are not consuming the calculated diet. In pasture-based systems, when provided with a choice, dairy cattle ingest the leafy portion of the plant and select green material over dead material [91]. Hence, the residual herbage mass will depend on the pasture type, pre-grazing herbage mass, and grazing pressure (time and stocking density). Additionally, the grazing efficiency, and therefore the amount of residual pasture, may also be affected by the genotype of the dairy cattle, with the New Zealand Holstein being better suited to pasture-based systems compared with the American Holstein and the milking regime, namely, once-a-day or more frequent milking [56,92,93,94,95,96,97].

Residual feedstuffs should be removed regularly, particularly when feeding high-moisture feeds, such as silage and potatoes, to minimize the risk of spoiling. In pasture-based systems, the residual herbage mass should ensure the future growth of the sward in order to maintain continual grazing.

Appetite is also affected by the position of the body of dairy cattle during feeding. It should be as similar to that adopted when they are grazing grass [9,54]. Cattle which ate with their heads in a similar position to when grazing produced more saliva, have higher intake and better rumination [9].

Availability of feedstuffs to cattle is affected by the frequency of feeding and access [10]. Increased feeding bouts and feed intake have been reported for feeding dairy cattle more than once in feedlot systems [10,98]. However, in pasture-based systems, provision of fresh breaks six times compared to two times per day did not increase the intake nor the milk production [87]. It is likely that the time of the day when the fresh break/paddock is offered and the total grazing time per day are more important for feed intake and milk production than number of fresh breaks offered per day [89]. During feeding, cattle push a proportion of the feed beyond reach. This should be regularly pushed back to maintain access and minimize overstretching and possible trauma [99]. Pushing the feedstuffs back into the bunker/feedpad may sometimes be enough to stimulate feeding activity anew [10,100,101]. Similarly, letting grazing dairy cattle on a ‘new’ break that has been previously incompletely grazed, is usually enough to stimulate feeding activity anew. In most pasture-based commercial settings, dairy cattle consume most of their daily allowance within 2–3 h from gaining access to a fresh break/paddock [87]. Ideally feeding facilities should provide 60 to 85 cm of space per cattle-head [54,84,102]. The dimensions vary depending on presence of headlocks/dividers and horns, age, breed, size and category of dairy cattle, and climate [9,54,84]. Due to reduced convective heat loss in crowded conditions, farms in hotter climates should provide a larger feeding space. Enough room for all cattle to feed at the same time is required for good food utilization and better production [9,102,103]. In pasture-based systems the stocking density on a paddock/break depends on the same factors as for feedlot-based systems in addition to the pasture/crop quality and quantity, amount of supplemental feed, antecedent experiences of each individual, and current environmental and social conditions [45,83,104,105,106,107,108,109,110,111]. This is important, as cattle are social animals and eat at the same time, often referred to as ‘social facilitation’ [9,10,112,113]. However, the social facilitation is somewhat less obvious in pasture-based automatic milking systems [83]. As feeding is affected by social ranking, younger and smaller dairy cattle, particularly heifers, are usually left aside if there is insufficient feeding space [10,54,101]. Similar behavior is usually seen in recent (less than a week) re-groupings [9].

Assessment of the appetite is usually subjective and is achieved by observing the feeding behavior of dairy cattle when fresh food is offered [9,112,114]. Proxy behavioral measures of hunger include time spent searching or acquiring food, rate of food intake, and rate of trade-offs between feeding and other activities in their time budgets [10,114]. Another measurement of appetite can be obtained by assessing prehension. Objective measures in commercial settings are difficult. Some assessment can be carried out by the use of individual feeding bins combined with video recording or electronic-identification systems, all of which are non-practical and/or expensive. On modern dairy farms, dairy cattle are kept and fed in groups; therefore, the individual feeding and recording is impractical and not a true representation. Video recording is also subjective, and it is time-consuming to assess the feeding behavior. The advantage of electronic systems is that the quality of the obtained information of some of them is very high [10,115]. Assessment of the appetite should also consider the dry matter intake per individual per day. Dry matter intake should average from 2.5% to 4.0% of the body weight, dependent on feed base, phenotype, milk yield, and stage of lactation [63,116]. In pasture-based systems, dry matter intake of up to 4.4% of the body weight at peak lactation have been achieved [86]. Lower values indicate that cattle are underfed or have a lowered appetite.

When assessing the appetite, the practitioner should always consider the accessibility of the food [5,7,9,16], degree of hunger [7,38], and palatability [7,9,75]. To maximize the intake of dry matter by healthy dairy cattle, feedstuffs should be prepared, stored, mixed, and delivered to the feeding area in ways that prevent contamination and spoilage [7]. Additionally, in pasture-based systems, the stage of maturation and energy density of the grass/crop are equally important in maximization of the intake.

### 5.2. Prehension

Prehension is the act of grasping the food and ability to drink with the mouth. It may be affected with disorders of the mouth cavity, nervous system, pharynx, and, rarely, esophagus and larynx. Prehension may also be impaired due to inability to swallow. It is important to differentiate between a depressed appetite and the inability to prehend food due to other causes (e.g., pain, paralysis). Dairy cattle with ad lib access to feed eat for 5.0 ± 2.5 h per day in a feedlot system (dependent on tie or free-stall system, diet, and physiological status of the dairy cattle individual of interest) and 7.0 ± 3.5 h in pasture-based systems (dependent on sward characteristics such as lush pasture or thorny bushes) in several eating bouts (4–20) [9,14,16,50,54,60,63,65,66,69,72,73,77,80,89,91,97,101,114,115,117,118,119,120,121,122,123,124,125,126,127,128,129,130,131,132,133,134,135,136]. Dairy cattle on pasture show a distinct diurnal feeding behavior [9,40,61,82,87,103,108,137]. They spend more time grazing during the day with rest and rumination around midday [45]. The diurnal feeding behavior in housed dairy cattle is less distinct or may be completely absent, particularly when fed on total mixed rations [10,100,129,138,139]. Longer grazing periods in late afternoons and early mornings are beneficial for cattle kept on pasture-based systems [58]. Incorrect milking management or insufficient pasture availability may result in extended grazing periods around midday. Grazing periods around midday should be shorter, particularly during hot days.

The average time spent eating and number of eating bouts depend largely on cow factors (e.g., age, stage and level of production, breed, social dominance), appetite, system of food delivery (e.g., feedlot or pasture-based), time of the day, and feeding related to other management practices (e.g., milking) [63,101,118,140]. The time spent on eating and number of eating bouts are heavily affected by the level of production, with high producers spending a longer time eating and, often, in more feeding bouts [54]. Additionally, the time spent eating is affected by the diet, e.g., grazing chicory and plantain requires more time on mastication at ingestion but less time on rumination compared to rye grass pasture [119,141]; diet composition and intake; fiber type and length; and age, size, breed, and production status of the dairy cattle individuals of interest [47,49,57,58,59,60,63,66,73,76,101,115,116,117,122,125,139,141,142,143,144,145]. In pasture-based systems, the time spent eating is a function of grazing time, biting rate, and the bite mass [96,109,146], which are dependent on the same factors as the residual herbage mass.

Heifers tend to eat less per feeding bout [7,64,118]. They prefer to visit the feeding facilities more frequently [6,126,129,147], probably due to their smaller rumen capacity [39,129]. Competition at the feeding platform/pasture is highest when dairy cows return from milking and when fresh food/a new pasture break/paddock is offered [9,10,54,83,90,100,101,102]. At these times, dominant dairy cattle demand priority for feeding and attempt to pick the high-quality food. Less dominant, and particularly submissive dairy cattle may have limited access to food at these times [6,7,8,9,10,54,66,101,112]. As these cattle eat less or choose to eat at times when there is less competition at the feeding platform, the available food may be of lower quality due to previous sorting by more dominant cows [5,6,9,10,63,90,131]. Sorting can be minimized by feeding a milled and properly mixed total mixed ration (TMR) diet. Aggression and competition when feeding on pasture is less common than in feeding barns, as grass is spatially distributed over large areas and all dairy cattle can feed at one time [148]. Grouping strategies can minimize the negative social interactions (e.g., avoiding grouping primiparous with multiparous cows or dairy cattle of different sizes or cows in different stages of the production cycle) [7,9,50,131]. Additionally, homogenous groups make management of nutrition easier, making it easier to formulate an appropriate ration or allowing better land use in pasture-based systems [54,131]. Dairy cattle are herd animals and eating in one individual stimulates the appetite in others, referred to as social facilitation [9,54,112,113]. Younger cattle learn to consume offered supplemental food or graze when exposed to experienced individuals than when learning to consume offered feedstuffs/grazing as a naïve group; this is referred to as social learning of feeding [105,146,149,150,151].

Assessment of prehension is usually carried out by observation or video recording. In practice, during the nutritional visit, the practitioner usually briefly assesses prehension by observing the acts of grasping, chewing, and swallowing in a several cows at the bunk/feedpad/pasture. These procedures are labor-intensive [98,145]. Video recording can be used as an alternative. Assessment of prehension can give useful information about the appetite, health, and diet quality.

Chewing movements during ingestion of feedstuffs, and/or time spent eating, have been incorporated into some automatic devices of recording cow behavior for the purposes of the detection of estrus or impending parturition and health [10,117,126,133,143,145,152,153]. Other electronic devices, such as electronic gates and geo-spatial identification readers, can be used to measure the time that individual dairy cattle spend at the bunk/feedpad/grazing [110,154,155].

### 5.3. Rumination

The act of chewing the cud (rumination) starts with an abdominal contraction followed by antiperistalsis of the esophagus from where the bolus (food bolus; cud) is delivered into the mouth [40,101,156,157,158,159]. In the mouth, the bolus is driven between the molar teeth by a single stroke of the jaw. Thereafter, chewing of the cud is carried out on one side of the mouth only, in a methodical grinding manner [156,157,158,159,160,161], and then re-swallowed [101,162]. Rumination is required to reduce the size of ingested particles in order to pass through the reticulo-omasal orifice, and also increases saliva production, which plays a role in buffering of the rumen fluid [48,59,67,68,79,101,122,141,155,161,162,163,164]. The rumen microbial degradation of ingesta hardly, if at all, influences the particle size, even when plant fiber is weakened by microbial fermentation. Chewing activity, particularly during rumination, is necessary to decrease the particle size. This increases the particle surface/volume ratio, and thus results in improved microbial access and rumen fermentation [101,141,155,162,165,166]. The act of re-chewing the food bolus is essential for its utilization by ruminants [16,59,67,79,122,161,164,167]. As the particle size decreases, the feed particles pass more readily and rapidly through the reticulo-omasal orifice (the critical particle size is assumed to be 1.18 mm) [133,168], the rumen fill decreases, and satiety receptors become inactive. Thus, rumination has a significant effect on the appetite of dairy cattle [48,101,162].

Rumination can be affected by diet composition and access, estrus, painful conditions, rumen movement dysfunction caused by metabolic or neurological conditions, rumen acidosis, and time budgets, in particular lying times. Factors affecting rumination such as individual dairy cattle signalment, climatic conditions, including heat stress and rain, day length, exercise, production status and level, stress, and time of day have been reviewed [2,21,38,39,40,41,49,126,129,159,162,169,170,171,172]. Time spent on rumination in dairy cattle depends on various factors, including feed quality and quantity (particularly the adequacy of fiber content and length), type of feeding, and body size, as well as on management factors such as the availability and quality of space for rest in a stress-free environment [9,47,49,57,59,63,64,68,69,73,76,117,155,156,159,160,161,162,164,173,174] and, in pasture-based systems, available grazing time [140]. Diets rich in fiber generally increase the chewing activity [18,59,66,73,83,123,141,142,175]. In contrast, diets rich in concentrate or roughage chopped to particles less than 1 cm in length reduce the chewing activity [9,49,58,60,116,142,160]. Unfortunately, fiber content alone is not a good predictor of the risk of ruminal acidosis [18,32,134]. Excessively long fiber particles can paradoxically increase the risk of acidosis [6]. Sorting of feedstuffs may result in both dominant and very submissive dairy cattle suffering from acidosis due to preferential concentrate uptake [6,7,32,36,72,175]. The theory of variable chewing time and effort being dependent on the fiber length alone is not consistently supported by research findings. Increased chewing during rumination has not always resulted in improved rumen fluid pH and decreased risk of subacute rumen acidosis [32,66,172]. Other factors that influence rumination time and chewing activity during rumination are the different rumen fermentation rates of various diets [58], reduced saliva production in cows that chew food at a faster rate when eating resulting in longer periods of no chewing activity [123], and rumen digestive potential, which influences the rumen pH and the volatile fatty acid composition [18,58,63,135,142]. Restricted feed availability, seen in many dairy feedlot systems, usually results in faster eating [54,172], swallowing of larger feed particles, and is associated with longer rumination times [6,47] but not always with a decreased risk of acidosis. In fact, the risk of acidosis may even be increased [32,172]. A comfortable and normal lying posture enhances rumination [9,13,131,139,164,176]. Most dairy cattle, during relaxed rumination, lie down with a slightly extended neck during the night and nearly 50% of all dairy cattle ruminate standing during the day [126,139,155,159,167,169]. Chewing movements stimulate production of saliva [6,18,21,48,49,58,66,71,72,75,101,122,134,160,161,164,167], lowering the risk of reduced fiber digestion, milk fat depression, displaced abomasum, fat cow syndrome, sub-acute ruminal acidosis (SARA), and associated conditions, including lameness, ruminitis, liver abscessation, metabolic acidosis, and caudal vena cava syndrome [5,6,14,18,21,32,33,34,35,37,41,49,53,58,66,67,122,133,142,164]. Dairy cattle lying down in a low-stress environment ruminate for a longer period [9] and this is associated with improved digestibility of the feedstuffs, feed conversion efficiency, and productivity.

Although lactating dairy cattle seldom ruminate over 10 h per day in total, cattle on a very rough diet may ruminate up to 12 h per day [101]. The maximum total chewing time has been estimated at 16 h per day [101]. Normal, healthy cattle ruminate for 7.0 ± 3.5 h a day in several rumination bouts (10 to 20) [9,14,21,48,54,59,63,66,69,72,76,77,89,114,117,119,121,122,123,124,126,127,128,129,130,132,133,134,135,139,142,143,159,162,163,164,169,173,176,177]. Each bout lasts 10–60 min (range 0.5–120.0 min) [76,101]. The length of each rumination bout varies with the availability of acceptable space, availability, composition, digestibility and type of the diet [47,59,75,76,101,178], interactions within a group [9,101,178], and social ranking. A diurnal pattern of rumination is seen in most dairy cattle [14,48,54,69,103,155,179], contrary to the diurnal pattern of appetite, meaning cattle ruminate when they are not prehending [45,103]. Additionally, a circadian pattern with most of the rumination occurring during night has been reported in dairy cattle in pasture-based systems [45,101,140].

The number of movements of the jaw required to chew a cud is usually indicative of the food quality. The number of chews is particularly, but not exclusively dependent on the fiber content [58,63,75,101]. The number of chews per cud also depends on the type, quality, and length of fiber particles [14,58,116,123]. Furthermore, the number of chews can be affected by many other factors [48]. For example, heifers and old cows with developing and/or malocclusive teeth usually make more chewing movements per cud to achieve the same grinding effect as cows of age 4–7 years [48]. Diets of acceptable quality should result in 60 ± 10 chewing movements per cud [47,48,59,76,117,139,144,163]. Less than 50 chewing movements per cud is indicative of insufficient fiber in the diet (e.g., lush pasture) [14]. A lower number of cud-chewing movements may also result from a significant stress or health problem [180]. Some studies have reported fewer than 50 chews per cud, but these are mainly older studies and investigated the effect of the addition of concentrate to the diet on milk production, with no concurrent measurements of rumen pH and rumen health assessment [76,144]. More than 70 chewing movements per cud is indicative of excessive fiber in the diet (e.g., low-quality straw) [1,5]. Generally, a higher number of chews per cud is not indicative of danger for the health of dairy cattle. However, the energy spent on chewing and associated depression in appetite results in decreased dry matter and energy intake and, therefore, lower milk production [59]. Less chews per cud are expected in pasture-based systems. Dairy cattle graze succulent plants in preference to drier, more mature plants [39]. Therefore, it is likely that less fibrous feed results in a reduced number of chews per cud. Increased fiber content and particle size usually results in increased chewing per unit of dry matter [73,101] and thus depressed appetite. Therefore, the total rumination time per day may remain unchanged [14,59,139]. This is important for assessments based only on total rumination time.

As rumination impacts the intake and utilization of the diet, it directly correlates to productivity and health [10,37,142,155,162,181]. Therefore, it is often used as a proxy to measure dairy cattle health and welfare [47,101,156,160,164,169,173,181,182]. For example, in a meta-analysis, decreased rumination was a good predictor of metritis but not subclinical ketosis [181]. Additionally, the rumination behavior can also be used as an indication of physiological changes such as calving or estrus [101,126,169,179,182]. Even slight changes in rumination (time spent on rumination or number of chews per cud) can be used as indicators of a potential subclinical problem with the diet, and actions can be taken before they become clinical [35,101,160,169]. Approximately 70–85% of cattle not eating, drinking, or sleeping at any one point in time should be ruminating [34,35,173]. Alternatively, in free-stall and pasture-based systems, it is recommended that at least 40% of all cows are ruminating at any given time [6,21,35,123,173]. This value may be affected by feed delivery and management events such as milking. Therefore, findings from a single observation of a population should be interpreted with caution. Continual recording of rumination by using electronic devices allows this parameter to be objectively measured. Dairy cattle not ruminating for prolonged periods when resting may have problems that should be investigated. Disturbed rumination at a population or group level may result from new social interactions following changes in the group composition or the presence of cows in estrus or from metabolic disorders such as ruminal acidosis [21,35,53,178]. Nutritional deficiencies or sub-optimal rumination can result in reduced immunity and increased susceptibility to infections [4,32,34,56]. For example, decreased rumination times in the first few weeks of the post-calving period were highly correlated with diagnosis of infectious disorders in the same period [156].

Rumination can be observed in real-time, by video recording or by using electronic automated systems [10,101,122,145,152,155,156,157,159,160,161,162]. Many of these electronic devices are capable of differentiating between bites and chews and are able to identify when the animal is ruminating or grazing [162,166]. The direct (visual) observation should be carried out by inspecting a number of individual dairy cows at a distance to avoid changing the cattle behavior or moving away and counting the number of chews per cud over a period of 5–10 min [167,183]. The practitioner should ensure an unobstructed view of the observed individual. Unfortunately, direct or video observation can only be carried out on a limited number of individual cattle due to time constraints [122,154,157,161,167]. Therefore, to ensure the collection of more robust data, automated systems have become a common practice on many advanced dairy farms. Automated systems are available in the form of collars [126,155,159,160,161,162,164,167,169,184], noseband pressure devices [117,122,143], other halter systems [133,134,145,167], rumen or vibration sensors [185], and activity monitors [157]. Another advantage of many automated systems is their objectivity [101,157]. One should be aware that, in many cases, the performance of these devices depends on the dairy system being used [157,160], age of the cattle [161], or other machinery/cattle/equipment-related factors [157,160].

### 5.4. Thirst

Thirst is the desire to drink water. It may be normal (eudipsia), increased (polydipsia), decreased (oligodipsia), or completely absent (adipsia). Drinking behavior and volume of water drunk are likely to affect both feed intake and production [186,187,188]. Drinking promotes further eating [186], which is particularly important for dairy cows for improved production. Feedstuffs with a higher water content result in decreased thirst at the population level. Conversely, feedstuffs with a high dry matter content result in an increased thirst. Dairy cattle tend to drink quickly, up to 20 L of water per minute [186,187]. Drinking from a large, calm surface elevated from the ground rather than from flowing water is preferred by dairy cattle [186,189,190]. Drinking from a bowl changes cattle behavior to more but shorter-duration drinking bouts per day [186].

Thirst and drinking behavior are affected by factors such as climatic conditions, diet composition, pregnancy, stage and level of production, and water quality [38,43,51,188,191,192]. The water intake is proportional to the increase in the ambient temperature [188]. Higher proportion of concentrates in the diet requires higher water intake [188]. Generally, lactating dairy cows drink 40–120 L/day and dry cows 17–70 L/day [187,189,193,194,195,196]. Additionally, thirst may be affected by the water availability, quality and quantity, access to the watering facility, and social dominance [9,51,186]. Dairy cattle should drink at intervals during the day [129]. On pasture, the number of drinking bouts and volume of water consumed may be affected by the distance between the grazing area and the water trough [186,193]. Water consumption in lactating cows is greatest at feeding and just after milking [71,186,187]. Whilst it may be disruptive to the cow flow, having water available at the exit of the milking shed as well as the feeding area may be beneficial [187]. Drinking behavior is also affected by social interactions [11,13,186,193]. Dairy cattle with a higher dominance in the group usually drink less frequently but larger volumes per drinking bout [186].

When assessing thirst, the availability and quality of the food and environmental temperature should also be considered. As dairy cattle eliminate a significant portion of their body heat through increased respiration, and exhale moisture rich air, higher environmental temperatures result in an increased thirst.

Unfortunately, the assessment of thirst is yet to be validated. As an interim measure, to assess the thirst and availability of drinking water at the facility, it is recommended that the practitioner observes for queuing and displacement at the water throughs within a short time after feeding [11,186]. Some measurements can be achieved by providing individual drinkers and automatic recording of individual cattle identity and water consumed [115]. Due to the cost, this system is unlikely to be widely used in the near future.

## 6. Physiological Parameters

### 6.1. Fecal Digestibility Scoring

The scoring of fecal digestibility is carried out by sieve washing (Table 4 [53,62,77,120,165]) and squeezing feces with a gloved hand (Table 5 (adapted from [1] with information from [1,5,7,18,20,35,53,62,76,77,165])). This can be carried out on a fresh dung pat or on feces collected from the rectum [5,124,197]. The authors of this article recommend squeezing feces from a fresh dung pat as this avoids unnecessary stress to the cattle and has been shown to be representative of fecal characteristics [20,191]. The presence of specific components in the feces may indicate where the problem in the feeding resides or the type of disorder of the digestive tract (Table 6 (adapted from [1] with information from [33,35,36,41,53,165,180])).

Feces may be screened by washing a cup of manure under running water through a sieve (0.2 to 0.3 mm) for approximately 30 s whilst being gently ‘massaged’ [1,5,77,124,197]. The assessment and interpretation with both techniques is similar. In the sieving technique, the remaining particles in the sieve are scored on a scale of 1 (nearly no particles left) to 5 (hardly any feces washed away). The residual material is used to qualify the digestion of consumed food (Table 4). Less than 10% of the starch should remain, and less than 12.5% of the screened dry matter should remain in the sieve. The presence of grain with residual starch indicates an insufficient preparation of the feedstuffs or impaired digestion [41,165] and is usually associated with a lower pH of feces due to hindgut fermentation [35,62]. The presence of feedstuffs longer than 15 mm usually reflects a lack of long fibers to maintain a healthy rumen mat or raft, a decrease in cud-chewing movements, and faster passage of ingesta through the digestive tract [41,53,77].

The fraction of the digested fiber and length of the particle size barely change after leaving the rumen [52,59,77,124,198]. Therefore, the measurement of the fiber/particle size in feces is representative of the particles leaving the rumen [59,77]. The most commonly recommended size of the particles in feces should be less than 1.18 mm, and the fibers should be short and fluffy [59,120,168]. However, the size of the particles in feces may be higher when the ration contains larger particles and ingestion is faster, such as in modern, high-producing cows [77,128,197]. The passage of ingesta through the digestive system in dairy cattle is influenced by factors such as age, weight and size, diet composition, digestibility, preparation, and feed intake [120,158,191]. Dairy cattle with a lower feed intake, such as dry cows and replacement heifers, and slower ingesta passage rate [41,62,128] have a smaller particle size. The slower passage rate probably allows for better fermentation, rumination, and digestibility of the diet [49,62,124].

Finally, fecal digestibility can be assessed by examination for fecal admixtures, which include mucin, undigested fiber, and grain (Table 6).

### 6.2. Fecal Scoring

The general health, state of rumen fermentation, and digestive function of cattle may be assessed by observation of the feces, particularly easily adopted to TMR feeding [20]. Fecal scoring is a tool that can be used in assessing the digestibility of the food, particularly the balance of digestible carbohydrates, fiber and protein, and the water intake. Freshly-deposited fecal pats should be assessed by observation, sliding the boot through the upper 1–2 cm of the pat [20], palpation of a handful with a gloved hand, and by sieve washing. The assessment of feces is usually easier on a concrete floor and difficult on deep pasture as the pat becomes deformed and sliding a boot may pose a challenge. These tests allow assessment of consistency and digestibility (Table 7 (adapted from [1] with information from [1,6,18,20,31,35,37,53,62])) on scale of 1 to 5. The scores are independent of the physiological status of the cattle of interest, their diet, and body size [57]. Due to changes in the diet and rumen function over lactation, the optimal fecal score varies. In general, cows in the first few days of lactation should have fecal score of 2.0–2.5, from 7 to 180 days in milk (DIM) 2.5–3.0, and for the remainder of lactation 3.0–3.5. Early-period dry cows should have a mean fecal score of 3.0–4.0 and the later dry period should be 2.5–3.5–4.0 [5]. Adverse changes in fecal scores may indicate problems with the balance of nutrients in the ration, inadequate mixing, sorting of food at the feeding area, and unacceptable competition at feeding.

Quality of feces in cattle on pasture-based systems is more variable, dependent on the content of easily digestible components and water. Feces produced by cattle consuming immature (lush) pasture tend to fall to the ground in shapeless deposits of a lower score (as low as a score of 2 without detriment). This score is usually non-detrimental to cows in active lactation [5]. Feces produced by cattle eating mature pasture, with increased structural non-digestible fiber, appears more solid (scores of 3 to 4).

### 6.3. Rumen Fill Scoring

Scoring of the rumen fill (Table 8 (adapted from [1] with information from [5,7,20,31,33,35,199])) is used to assess feed intake [161,199] and the speed with which food is moving through the digestive tract [20]. The degree of fill is a function of the diet composition, feed intake, the fermentation speed, and the rumen outflow rate [161,199]. This scoring system has been validated as relatively robust and repeatable [161]. Some apparent fullness of the rumen can be artefactual due to space-occupying conditions (e.g., large uterus in late pregnancy resulting in high rumen fill scores) or medical conditions such as bloat or vagus indigestion.

Rumen fill scoring is a visual assessment carried out from behind and slightly to the left of the animal. The rumen is assessed by observing the left sublumbar fossa and flank of the dairy cattle individual of interest [5,20,31,199].

The assessment should be carried out after a minimum of 1.5 h following their first feed for the day. If performed sooner, artificially low rumen scores may be recorded. The assessment should be carried out when there is no visible contraction of the rumen with the dairy cattle individual of interest standing with all four feet on the same level [199].

Note: The rumen fill described in this text ignores that the rumen may be bloated (meaning a score > 5) when urgent intervention is required.

## 7. Conclusions

The practitioner should use a systematic holistic approach to obtain an accurate profile of abnormal findings at the individual, group, and herd level. This includes the use of cow signs in addition to the history, feed analysis, environmental observations, production key performance indicators (KPIs), management information, metabolic profiles, and health records. Once obtained, the profile can be used to identify problem areas and combine the physical findings with assessment of behavior and health of cattle. Information gathered through assessment of cow signs should be used as an advisory tool to assist producers in decision making about their management practices and farm facilities. The assessment should be carried out in the presence of the producer and all limitations should be considered. False interpretation of ‘cow signs’ can lead to incorrect troubleshooting decisions. All problematic areas should be listed and prioritized by the importance and the ease of implementation of the corrective changes, including a cost–benefit analysis. Increasing a producer’s awareness of the value of auditing important cow signs will increase their participation and improve outcomes.

## Figures and Tables

**Table 1 animals-12-01352-t001:** Scoring of obtundancy in cattle.

Score	Description	Head and Body	Reaction				
			General	Herd-Mates	Audio and Visual Stimuli	Surrounding	Approaching Person
1	Normal	Head up	Bright and alert	Interacts	Prompt reaction	Prompt reaction	Readily moves away from examiner or tries to make contact with nose/tongue
2	Mild obtundancy	Head up	Slower but still bright and alert	Avoids active interaction	Ignores mild stimuli	Decreased responsiveness	Moves away from examiner slower than normal. Rarely tries to make contact with its nose/tongue
3	Moderate obtundancy (dull)	Head down, ears drooped, no rumen fill (may appear floppy)	Sleepy. Sometimes walks into objects	No interaction with herd mates	Responsive only to very loud and painful stimuli and vigorous handling	Low responsiveness	Moves away from examiner very slowly
4	Severe obtundancy (stupor)	Head down. Abdomen gaunt. Sometimes recumbent	Only reacts to prolonged noxious stimuli	No reaction	Blunted responsiveness only to very noxious stimuli	Low responsiveness	Unresponsive
5	Coma	Recumbent	No reaction even to noxious stimuli	No reaction	No responsiveness to examiner or surrounding	Unresponsive	Unresponsive

**Table 2 animals-12-01352-t002:** Scoring of hyperexcitability in cattle.

Score	Description	Head and Body	Reaction				
			General	Herd-Mates	Audio and Visual Stimuli	Surrounding	Approaching Person
1	Normal	Head up	Bright and alert	Interacts	Prompt reaction	Prompt reaction	Readily moves away from examiner or tries to make contact with nose/tongue
2	Hyperreactive	Head up	Bright and alert	Active interaction	Reacts faster or stronger than usual	Increased responsiveness	Moves away from examiner faster than normal. Rarely tries to make contact with its nose/tongue
3	Restless	Head up, ears often erect	Overly alert	Mildly disturbs other members of the group	Vigorous response to mild stimuli	Vigorous response	Moves away from or towards examiner quickly
4	Mania	Head up, ears erect	Restless	Severely disturbs other members of the group	Over-reaction to minor stimuli	Very vigorous response	Runs away or towards examiner
5	Frenzy	Head up, ears erect and more caudally positioned	Restlessness and constant movement	Attacks other members of the group	Severe reaction to minor stimuli to no response	Unprovoked aggression towards inanimate objects	Trying to attack examiner

**Table 3 animals-12-01352-t003:** Scoring of the fecal soiling of the perineal area in cattle. The percentage (in the brackets) refers to the proportion of the perineal area that is stained. Perineal area in this table refers to area around the anus, caudal hindlimb and rump, and tail.

Score	Description
1	No fecal perineal staining
2	Mild; Few flecks of perineal staining (2–10%)
3	Moderate; Maximum up to 30 of the perineal area stained with feces (11–30%)
4	Severe; Large portion of the perineal area stained with feces (31–60%)
5	Very severe; Nearly whole perineal area stained with feces (>60%)

**Table 4 animals-12-01352-t004:** Scoring of the digestive function by sieving for 30 s with a gentle ‘massaging’.

Score	Description	Reasons and Interpretation
1	0–25% of the original volume left after sieving Fiber left in the sieve of short length and fluffy (<0.5 cm)	Excellent fiber digestion Ideal score
2	26–35% of the original volume left after sieving Fiber left in the sieve mainly of short length (<0.5 to 1 cm) Some larger, undigested fiber particles detectable	Slightly impaired digestion Less than ideal food quality Slightly impaired rumination Common in lactating and dry cows
3	36–50% of original volume left after sieving Some fiber left in the sieve > 1 cm long	Poor digestion Problems with processing the grain (not broken) Poor formation of rumen mat Not acceptable for lactating cows May be acceptable for dry cows and heifers due to slower passage time
4	51–75% of the original volume left after sieving Bigger undigested food particles Fiber particles sometimes >2 cm long	Poor digestion Poor formation of rumen mat Poor rumination Forages of poor quality Not acceptable for any class of dairy cattle May indicate acidosis
5	Less than 10–15% reduction after sieving Bigger food particles Rough fiber particles often >2 cm long Undigested components of the feed ration are clearly recognizable Casts of intestinal mucosa and fibrin may be present	Very poor digestion No formation of rumen mat Very poor rumination Forages of very poor quality Not acceptable for any class of dairy cattle May indicate acidosis or enteritis

**Table 5 animals-12-01352-t005:** Scoring of the digestive function by squeezing feces with a gloved hand.

Score	Description	Reasons and Interpretation
1	Creamy homogenous emulsion No visible undigested food particles Shiny surface of fresh feces	Good passage of ingesta through the digestive tract Good digestion Good food quality Good rumination Ideal score for cattle
2	Creamy homogenous emulsion Few undigested food particles of small size Shiny surface of fresh feces	Slightly impaired passage of ingesta through the digestive tract Slightly impaired digestion Less than ideal food quality Slightly impaired rumination Common in lactating and dry cows
3	Feces not homogeneous. Some undigested particles On hand squeeze some undigested fibers stick to the fingers Dull to shiny surface of fresh feces	Higher than normal speed of passage of the ingesta through the digestive tract Poor formation of rumen mat Poor digestion Problems with processing the grain (not broken) Acceptable score for dry cows and heifers fed on a high roughage diet due to slower passage rate
4	Bigger undigested food particles After squeezing a ball of undigested food remains in the hand Particles sometimes >2 cm Dull surface of fresh feces	Higher than normal speed of passage of the ingesta through the digestive tract Poor formation of rumen mat Poor digestion Forages of poor quality Poor rumination Gastro-intestinal parasitism
5	Bigger food particles Undigested components of the feed ration are clearly recognizable Very dull surface of fresh feces	High speed of passage of the ingesta through the digestive tract Poor formation of rumen mat Poor digestion Forages of very poor quality Very poor rumination

**Table 6 animals-12-01352-t006:** Admixtures of feces and common reasons related to nutrition, excluding pathologic conditions associated with various infectious disorders.

Component in the Feces	Reasons
Undigested fiber	Higher than normal speed of passage of the ingesta through the digestive tract Poor formation of rumen mat Poor digestion or rumen fermentation Forages of poor quality Poor rumination
Undigested grain	Higher than normal speed of passage of the ingesta through the digestive tract Poor formation of rumen mat Poor digestion, particularly acidosis Problems with processing the grain (not broken) Very dry silage Slug feeding NOTE: often husk only present and starch digested—careful assessment required
Mucin	Acidosis Increase in the digestive role of the hindgut Excessive acid production in the hindgut
Bubbly diarrhea	Excess in fermentable carbohydrate, particularly compared to fiber content—often characterized by putrid smell Acidosis (due to hind gut fermentation of carbohydrates and formation of gas)—often characterized by acidic smell
Variability within the group of cattle	Feedstuffs not mixed well Parts of food moldy

**Table 7 animals-12-01352-t007:** Description of the score system of quality of feces, common nutritional causes, interpretations of the findings, and actions to be taken.

Score	Description	Causes	Notes/Action to Take
1	Very liquid Watery Thin Runs through fingers of gloved hand Diarrhea Undesirable score	Various disorders of digestive tract Various generalized disorders Gastro-intestinal parasitism Excess of an osmotic gradient in the intestine Excess readily fermentable carbohydrates Lack of fiber Some mineral excess or poisonings Moldy feed Acidosis (lighter color and low pH; usually presence of bubbles due to fermenting starch) Hindgut fermentation Very short passage time of ingesta	Call veterinary practitioner Treat the reason for diarrhea Treat the dehydration
2	Runny; custard-like consistency Does not form a distinct pile Splatters moderately when hits the ground or concrete Pat measures less than 2.5 cm in height More watery than optimal	Cattle on lush pasture Gastro-intestinal parasitism Excess readily fermentable carbohydrate Lack of functional fiber Excessive intake of sand/soil	If single patient—monitor only If multiple patients—re-evaluate the diet
3	Porridge-like appearance with several concentric rings, a small depression or dimple in the middle Makes a plopping sound when hits concrete floors Spreads slightly on impact and settling Feces pat measures up 4 to 5 cm Sticks to the shoes	Cattle on lush pasture Optimal level of total and functional fiber	Ideal score for lactating cows If in dry stock and replacements—re-evaluate the diet (optimal diet should result in drier feces)
4	Thick porridge like consistency Feces pat measures up over 5 cm Original form very slightly distorted on impact and settling Firmly sticks to the shoes when touched Concentric rings evident	The level of total and functional fiber is high Low salt Low water Low protein and/or starch Adding extra grain and/or protein to the diet can decrease the score	Ideal score for dry stock and replacements If in lactating cows—re-evaluate the diet
5	Appears as firm fecal balls Original form not distorted on impact and settling Resembles horse feces Undesirable score	Excess of fiber (e.g., straw-based diet) Lack of rumen available starch Lack or rumen available protein/urea Dehydration (e.g., water deprivation) Blockage of digestive tract	Call veterinary practitioner Re-evaluate the diet

**Table 8 animals-12-01352-t008:** Description of the scoring system of rumen fill, with causes and interpretations of the findings.

Score	Description	Causes and Interpretation
1	A deep dip in the left flank. More than one hand-width deep Rectangular appearance The skin under the lumbar vertebrae curves inwards. The skin fold from the hook bone goes vertically downwards	Cattle have eaten little or nothing Sudden illness Insufficient food Unpalatable food Alarming situation
2	The skin under the lumbar vertebrae curves inwards for a hand width behind the last rib Triangular appearance (referred to as ‘danger triangle’) The skin fold from the hook bone runs diagonally forward towards the last rib The paralumbar fossa behind the last rib is one hand-width deep	Common in cattle in the first week after calving In other cattle is alarming situation May be indicative of acidosis Later in lactation sign of Insufficient food intake Too fast passage of food
3	The skin under the lumbar vertebrae goes vertically down for less than one hand-width and then curves outward The skin fold from the hook bone is not visible. The paralumbar fossa behind the last rib is still just visible	Correct score for lactating cows and beef cattle on pasture Good food intake Good timing of passage of food
4	The skin under the lumbar vertebrae curves outwards No paralumbar fossa is visible behind the last rib	Correct score for cows in late lactation Correct score for beef cattle in feedlot Correct score for early dry cows
5	The lumbar vertebrae are not visible as the rumen is very well filled The skin over the whole belly is quite tight No visible transition between the flank and ribs No visible transition between the flank and transverse processes	Correct score for late dry cows Correct score for heifers

## Data Availability

Not applicable.

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
