# Peer review of "The Value of ‘Cow Signs’ in the Assessment of the Quality of Nutrition on Dairy Farms"

_animals, 2022, doi:10.3390/ani12111352_

Round 1
Reviewer 1 Report
This review manuscript is very interesting and something news in area cows nutrition, milk production, manangment, walfare and especially of control of health in dairy herds.
In my opinion Part 2. Litererature review and search terms should be deleted, because it is not necessary as explanation what parts in manuscript missing.
In part 6. Physiological parameters should indicated as 6.1 Rumen fiil scoring and 6.2.Fecal digestibility scoring (should changes places),
Literature data and references should precise checking in whole texts expecially in tables.
This manuscript is acceptable for publicitation in Animals after minor corrections.
Author Response
Thanks to the reviewer for the constructive criticism.
Part 2 deleted.
We kept part 6 in alphabetical order, trying not to indicate that one physiological parameter is more important than other.
All referencing checked again.
Reviewer 2 Report
The article addresses important questions about behavioral and physiological parameters for evaluating nutrition in dairy cows. The only restriction to mention is about references, most of them are more than 5 years old. The article must be accepted with minor corrections, described below:
- line 87: insert a dot;
- line 130: add the term "... Table 2)" in the previous line;
- lines 544 and 589: see error message;
- there is no call in the text to Table 7.
Author Response
Thanks to the reviewer for the constructive criticism.
Line 87 full stop inserted.
Placement of text (see Table 1 and Table 2) changed to the previous line.
Error messages may have occurred due to different systems used. In our system (and it is my understanding of the journal system) error messages are not seen.
Table 7 is referred to in line 590.
Reviewer 3 Report
In dairy herds, the use of cow signals as a management tool has increased in importance in recent years, but there is still much that can be applied from this body of knowledge for a more efficient dairy production that safeguards the health and well-being of animals.
The publication of Jan Hulsen's "Cow Signals: A Practical Guide for Dairy Farm Management" in 2006 is an important milestone in this topic, but despite the obvious usefulness of this guide there is still a great deal of room for progress on this topic.
This manuscript follows in this direction, focusing on the application of this knowledge in nutritional management, which obviously can be very useful for those who carry out their professional activity in the production of milk. Also, I do not question the usefulness of this review in the form of a guide as it is very comprehensive, with a solid grounding on the usefulness of the various evaluated parameters.
However, if this review is to be useful for “farm advisors, consultants, nutritionists, practitioners and dairy farmers", it will perhaps have to be more straightforward and less exhaustive, avoiding being basically a compilation of "cow signs" related to nutrition. In short, all the information contained in this review can be very useful but applying it to the day-to-day of a dairy farm may be unfeasible as it is very time-consuming.
Currently, in most dairy farms, individual production is easily accessible, obviously constituting a very effective indicator of the level of animal intake. Other parameters that are also usually determined individually, such milk fat, can also be useful in this regard. In short, assessing some of the referred “cow signs” can be redundant in dairy farms with a minimum of technology. Therefore, I recommend that the authors adapt this system to these dairy farms, to simplify it.
Another aspect where this review fails is that it does not adapt the “cow signs" system to the different phases of the dairy cycle. For example, the interpretation or scoring of some parameters can vary greatly depending on whether you are evaluating a dry cow or a cow at peak production. The relevance of some parameters also varies throughout the phases of the dairy cycle, as well as the animals' nutrition itself also changes, and this is not consistently addressed in this review.
I believe the authors have extensive experience in this area and, therefore, any additional critical input from them would be most welcome to improve this manuscript so e.g. that it can be more easily applicable in the field.
Author Response
Thanks to the reviewer for the constructive criticism.
We have added a new section: Limitations of the 'cow signs' that addresses all of the reviewer's comments.
Additionally, we have changed one sentences in the conclusion to: "The assessment should be carried out in the presence of the producer and all limitations should be considered. False interpretation of ‘cow signs’ can lead to incorrect troubleshooting decisions."
Reviewer 4 Report
Dear Authors,
The present study is a review that aims to provide dairy farm advisors, consultants, nutritionists and practitioners and their dairy farmer clients with an additional toolkit that can be used in the assessment of the quality of their dairy cattle nutrition. The assessment of the quality of the nutrition is based, according to the authors, on the detection of various “cow signs”. These cow signs are behavioral, physiological and management parameters that can be observed and measured.
The main advantage of this assessment is that the observation and interpretation of these cow signs do not require specialized equipment, laboratory analyses or sophisticated software, so they can be used as a field approach to evaluate the quality of nutrition. From this point of view, the present study is of particular interest. The main disadvantage of the cow signs is that they cannot be used as a sole tool for assessment of the quality of nutrition and they also require experience from those who “read” them.
The review is well written, and the use of English is excellent. Regarding the structure of the manuscript, I must note that the abstract is concise and reflects the main idea of the study. The introduction provides sufficient background information for the readers, either they are specialists or non-specialists, something that is important for review articles. The chapters used by the authors are comprehensive, enriched with many bibliographic references and accompanied by tables which provide a lot of information in a clear and illustrative way. I would expect, however, that they would have a different structure. According to the introductory sentence of the simple summary the cow signs are behavioral, physiological and management parameters. In this sense, it might be preferable for the authors to adopt this kind of structure with the chapters, using subchapters where is necessary. The conclusion is clear, concise, and fully corresponds to what was written in the main text. Finally, the literature is extensive, informative, and well targeted. I only noticed 2 references inside the main text that need to be cited (L544, L589).
Ending I would like to express the opinion that it would be useful if the authors listed at some point in the text a subchapter in which the weaknesses of the proposed assessment of the quality of nutrition through the cow signs would be mentioned. As the study shows, the correct detection and interpretation of the cow signs require experience, as the quality of the nutrition depends on many factors. False interpretation of cow signs can lead to incorrect troubleshooting decisions. For this reason, I consider it necessary for the authors to state clearly, albeit briefly, the main weaknesses and limitations of the proposed practical evaluation of the quality of nutrition in dairy cattle.
Author Response
Thanks to the reviewer for the constructive criticism.
We have added a new section: Limitations of the 'cow signs' that addresses all reviewer's concerns. Additionally, we have changed one sentence in the conclusion that now read: "The assessment should be carried out in the presence of the producer and all limitations should be considered. False interpretation of ‘cow signs’ can lead to incorrect troubleshooting decisions."
References in L544 and 589 have occurred due to failure of the PDF conversion. Hope now fixed.
Round 2
Reviewer 3 Report
The only significant change in this review, regarding the first submitted version of the manuscript, was the inclusion of paragraph "2. Limitations of 'cow signs'", with many other suggested amendments not being considered by the authors.
By the way, I warn to the possible contradiction in these two sentences:
- “Observation of cow signs does not require specialized equipment, laboratory analyses and investment in significant ineffective labor 66 time” lines 65-67
- “Secondly, on many dairy enterprises, a significant impediment to implementation may be the time demands to correctly identify and utilize the ‘cow signs’.” Lines 93-94
I don't question the usefulness of this review but, in my opinion, it misses a chance to improve its applicability in the field in the current dairy production context. Given the individual emphasis of this review (and not on the herd) it will perhaps be more useful e.g. to prepare future buiatricians for the clinical examination of the digestive system in cattle in the event of possible disease
Author Response
Thanks to the reviewer for the constructive criticism.
Lines 65-67 corrected.
Additional material added to address the last paragraph of the reviewers comments:
The discussion in this review emphasizes individual cattle observations but ‘acceptable norms’ at population level are provided where applicable. We hope that this material can be also used for training of future practitioners in examination of the alimentary system. (L104 - 107)